# Neoadjuvant Chemotherapy of Triple-Negative Breast Cancer: Evaluation of Early Clinical Response, Pathological Complete Response Rates, and Addition of Platinum Salts Benefit Based on Real-World Evidence

**DOI:** 10.3390/cancers13071586

**Published:** 2021-03-30

**Authors:** Milos Holanek, Iveta Selingerova, Ondrej Bilek, Tomas Kazda, Pavel Fabian, Lenka Foretova, Maria Zvarikova, Radka Obermannova, Ivana Kolouskova, Oldrich Coufal, Katarina Petrakova, Marek Svoboda, Alexandr Poprach

**Affiliations:** 1Department of Comprehensive Cancer Care, Masaryk Memorial Cancer Institute, 656 53 Brno, Czech Republic; holanek@mou.cz (M.H.); bilek@mou.cz (O.B.); zvarikova@mou.cz (M.Z.); obermannova@mou.cz (R.O.); petrakova@mou.cz (K.P.); msvoboda@mou.cz (M.S.); poprach@mou.cz (A.P.); 2Department of Comprehensive Cancer Care, Faculty of Medicine, Masaryk University, 625 00 Brno, Czech Republic; 3Research Centre for Applied Molecular Oncology, Masaryk Memorial Cancer Institute, 656 53 Brno, Czech Republic; 4Department of Pharmacology, Faculty of Medicine, Masaryk University, 625 00 Brno, Czech Republic; 5Department of Radiation Oncology, Masaryk Memorial Cancer Institute, 656 53 Brno, Czech Republic; tomas.kazda@mou.cz; 6Department of Radiation Oncology, Faculty of Medicine, Masaryk University, 625 00 Brno, Czech Republic; 7Department of Oncological Pathology, Masaryk Memorial Cancer Institute, 656 53 Brno, Czech Republic; fabian@mou.cz; 8Department of Cancer Epidemiology and Genetics, Masaryk Memorial Cancer Institute, 656 53 Brno, Czech Republic; foretova@mou.cz; 9Faculty of Medicine, Masaryk University, 625 00 Brno, Czech Republic; 466944@mail.muni.cz; 10Department of Surgical Oncology, Masaryk Memorial Cancer Institute, 656 53 Brno, Czech Republic; coufal@mou.cz; 11Department of Surgical Oncology, Faculty of Medicine, Masaryk University, 625 00 Brno, Czech Republic

**Keywords:** triple-negative breast cancer, neoadjuvant chemotherapy, early clinical response, pathological complete response, *brca* mutation, platinum salts

## Abstract

**Simple Summary:**

Neoadjuvant chemotherapy (NACT) is the standard treatment for early-stage triple-negative breast cancer (TNBC). Achieving pathological complete response (pCR) is considered an essential prognostic factor with favorable long-term outcomes. The administration of NACT regimens with platinum salts is associated with a higher pCR rate. However, with unclear treatment guidelines and at the expense of a higher incidence of adverse events. Identifying patients and circumstances in which the benefits of platinum NACT outweigh inconveniences is still an ongoing challenge. Considering early clinical response (ECR) after the initial standard NACT cycles together with other suitable predictors could be useful to decide about the administration of platinum salts in clinical practice. The results of this large single institutional retrospective study of consecutive patients showed the significant role of adding platinum salts in older patients with high-proliferative early responded tumors and persisted lymph nodes involvement regardless of *BRCA1/2* status.

**Abstract:**

Pathological complete response (pCR) achievement is undoubtedly the essential goal of neoadjuvant therapy for breast cancer, directly affecting survival endpoints. This retrospective study of 237 triple-negative breast cancer (TNBC) patients with a median follow-up of 36 months evaluated the role of adding platinum salts into standard neoadjuvant chemotherapy (NACT). After the initial four standard NACT cycles, early clinical response (ECR) was assessed and used to identify tumors and patients generally sensitive to NACT. *BRCA1/2* mutation, smaller unifocal tumors, and Ki-67 ≥ 65% were independent predictors of ECR. The total pCR rate was 41%, the achievement of pCR was strongly associated with ECR (OR = 15.1, *p* < 0.001). According to multivariable analysis, the significant benefit of platinum NACT was observed in early responders ≥45 years, Ki-67 ≥ 65% and persisted lymph node involvement regardless of *BRCA1/2* status. Early responders with pCR had a longer time to death (HR = 0.28, *p* < 0.001) and relapse (HR = 0.26, *p* < 0.001). The pCR was achieved in only 7% of non-responders. However, platinum salts favored non-responders’ survival outcomes without statistical significance. Toxicity was significantly often observed in patients with platinum NACT (*p* = 0.003) but not for grade 3/4 (*p* = 0.155). These results based on real-world evidence point to the usability of ECR in NACT management, especially focusing on the benefit of platinum salts.

## 1. Introduction

Despite current progress in the oncology treatment, triple-negative breast cancer (TNBC) remains a highly aggressive disease with a significantly shorter overall survival compared with other breast cancer subtypes with five-year mortality reaching 40% [1,2]. For the needs of daily routine clinical practice, TNBC is defined by the lack of estrogen receptor (ER), progesterone receptor (PR), and human epidermal growth factor receptor 2 (HER2) [3,4]. Despite biological heterogeneity according to gene expression patterns [5,6,7], neoadjuvant therapy of TNBC is usually based on conventional chemotherapy. Administration of new treatment strategies, such as immune-checkpoint inhibitors [8,9] and poly-adenosine diphosphate ribose polymerase (PARP) inhibitors [10,11], is often limited to metastatic TNBC and depends on the reimbursements from health care insurance companies. Nevertheless, immunotherapy has a promising effect also in the neoadjuvant setting [12].

Neoadjuvant chemotherapy (NACT) is the standard and preferred treatment option for TNBC of stage II–III [13,14,15,16]. Achievement of pathological complete response (pCR) after NACT, characterized as the complete disappearance of invasive carcinoma from breast and axillary lymph nodes, is one of the main treatment goals and is associated with a lower risk of recurrence and death [17,18,19,20,21]. Therefore, the ability to identify patients with the greatest benefit from NACT and administer chemotherapeutic regimens with the highest probability of pCR achievement is crucial. Several predictive and prognostic markers, such as age, menopausal status, tumor grade, the value of Ki-67 proliferative index, presence of tumor-infiltrating lymphocytes (TILs), *BRCA1/2* status, and programmed death-ligand 1 (PDL1) expression, could be used in clinical practice [22,23].

The effectiveness of NACT may already be assessed during neoadjuvant therapy by determining early clinical response (ECR). The previously published studies evaluated ECR after the initial two cycles o NACT, wherein favorable response was associated with a higher likelihood of pCR achievement or better long-term treatment outcomes [24,25,26].

NACT of TNBC is usually based on anthracyclines and taxanes [13,14]. The benefit of other agents, such as platinum salts (e.g., carboplatin, CBDCA and cisplatin, CDDP), was repeatedly discussed [27,28,29,30,31]. Platinum salts have a different spectrum of antitumor efficacy. They are transported into a tumor cell and form highly reactive molecules, which have a similar effect to alkylating cytostatics. However, they do not directly alkylate deoxyribonucleic acid (DNA). They bind to DNA, form intercalating cross-link DNA strands, and prevent replication, which leads to the interruption of replication and apoptosis. Platinum salts should be more effective in TNBC due to more frequent damage of the repair system of DNA breaks caused by germline mutations in tumor suppressor genes (*BRCA1/2*, *PALB*) or by somatic mutation leading to homologous recombination deficiency [32,33,34,35].

Following the ASCO neoadjuvant therapy guideline for breast cancer [36], the addition of platinum salts may be offered to selected high-risk TNBC patients to increase the likelihood of pCR. Based on a personalized approach, the physician needs to balance potential benefits and harms. However, clear recommendations for which patients platinum salts are suitable are not available. The consequences of expected more frequent pCR in platinum regimes are much less certain in terms of long-term outcomes.

The aim of this retrospective study of consecutive early TNBC patients treated neoadjuvantly outside of clinical trials was an efficacy evaluation of NACT involving platinum salts compared to standard regimens and identifying patients who benefit most from platinum salts adding. The effect of NACT evaluated after initial four cycles of standard regimens expressed by ECR was used together with known clinical characteristics to predict pCR in the context of platinum salts. The output is recommendations for platinum salts administration with a potential impact on clinical practice. In addition, the long-term treatment outcomes were assessed.

## 2. Materials and Methods

### 2.1. Patient Population and Follow-Up

Patients with histologically confirmed, nonmetastatic breast cancer, who underwent surgery at MMCI between 2012 and 2019 and previously treated with NACT were screened for eligibility for this study. A total of 848 patients were included. Patients with synchronous breast cancer, with unknown subtype, treated with hormonal neoadjuvant treatment, treated within clinical trials, and male breast cancers were excluded. From this cohort (785 patients), patients with TNBC (*n* = 243) were selected. The final analyzed group consisted of 237 patients (Figure 1). All patients provided written informed consent with the processing of tissue samples for research purposes, and the study was allowed by the local ethics committee. Patient follow-up during and after NACT was based on established standards of care in our institution and on international guidelines [14]. Follow-up included imaging and clinical examination; blood tests and additional imaging was performed in some cases (e.g., suspicion of disease relapse) based on attending physician discretion. The follow-up schedule was as follows: In the first two years after 3–4 months, in the next three years every six months, and then once a year. Laboratory parameters, including neutrophil count, lymphocyte count, platelet count, monocyte count, hemoglobin level, plasma lactate dehydrogenase (LDH) level, plasma C-reactive protein (CRP) level, plasma albumin level, serum CEA level, serum CA 15-3 level, were obtained from the blood samples taken at the beginning of NACT (before the first cycle of NACT). Inflammation-based prognostic scores were established, namely the neutrophil to lymphocyte ratio (NLR), lymphocyte to monocyte ratio (LMR), systemic immune-inflammation index (SII, neutrophil × platelet/lymphocyte), and C-reactive protein to albumin ratio (CAR).

### 2.2. Pathological Assessment, Breast Cancer Subtypes, and BRCA1/2 Germline Mutation Testing

Histopathological data were obtained from original pathological reports. Tumor grade, expression of ER and PR, HER2 status, and value of proliferative index Ki-67 were assessed as follows: Grading was determined by the Nottingham Histologic Score system in accordance with the WHO classification of breast tumors [37]; all immunohistochemical assays were performed using the Ventana Ultra Benchmark immunostainer (Roche Diagnostics, Basel, Switzerland). Monoclonal antibodies were used as follows: ER-SP 1 (Ventana/Roche); PR-NCL-PGR-312 (Novocastra/Leica Biosystems, Nussloch, Germany); Ki-67-30-9 (Ventana/Roche); HER2- 4B5 (Ventana/Roche). HER2 status was determined in accordance with the currently valid ASCO/CAP guidelines [4]. For this study, patients with negative or low expression (≤10%) for ER and PR and negative for HER2 were also classified as having TNBC [38]. Ki-67 proliferative index was assessed in a 20× field with the highest proliferative activity (hot spot) by QuPath software [39]. The proliferative index less than or equal to 20% was classified as low, 21–40% as intermediate, and >40% as high. The arbitrarily chosen cut-off value of 65% was considered to determine very high proliferation with regard to multiple endpoints and Ki-67 distribution.

Genomic DNA required for the identification of germline *BRCA1/2* mutations was isolated from blood samples. High-resolution melting (HRM) curve analysis [40] followed by Sanger sequencing was used for *BRCA1/2* germline mutation testing since 2007. Multiplex ligation-dependent probe amplification (MRC Holland) has been used for the detection of large rearrangements since 2005. Next-generation sequencing methods (NGS) [41] have been used since 2014. NGS started with the TruSight Cancer panel (Illumina, San Diego, CA, USA). NimbleGen SeqCap EZ Choice (Roche) has been used since 2016 for the creation of a sequencing library with the use of a multi-gene (226 genes) panel called CZECANCA (CZEch CAncer paNel for Clinical Application) [42]. Variants were classified using the five-class system and results were interpreted following the recommendations of the Association for Molecular Pathology, American Society of Clinical Oncology, and College of American Pathologists [43]. *BRCA1/2* germline genetic testing was considered in TNBC patients following the guidelines for high-risk patients [44].

### 2.3. Chemotherapy Regimens

Patients were treated with standard neoadjuvant anthracyclines(A)-based regimens (AC, doxorubicin 60 mg/m^2^ plus cyclophosphamide 600 mg/m^2^ every 3 weeks or dose-dense (DD) administration for four cycles every two weeks; EC, epirubicin 100 mg/m^2^ plus cyclophosphamide 830 mg/m^2^ every three weeks for four cycles; FEC100, 5-fluorouracil 500 mg/m^2^ plus epirubicin 100 mg/m^2^ plus cyclophosphamide 500 mg/m^2^ every three weeks for three cycles) followed by taxane (T)-(P, paclitaxel 80 mg/m^2^ weekly for 12 cycles or D, docetaxel 75–100 mg/m^2^ every three weeks for three or four cycles), T-based only regimens (e.g., TC, docetaxel 75 mg/m^2^ plus cyclophosphamide 600 mg/m^2^ every three weeks for six cycles), or CMF (cyclophosphamide 600 mg/m^2^, methotrexate 40 mg/m^2^ and 5-fluorouracil 600 mg/m^2^ every three weeks for six cycles). Platinum salts were administered after A-based regimens; CDDP monotherapy (75 mg/m^2^ every three weeks for three or four cycles), CBDCA combined with T (AUC 1.5–2 plus paclitaxel weekly every week for 12 cycles). Administration of platinum salts was chosen by an attending physician after initial four cycles of NACT, taking into account especially *BRCA1/2* germline mutation or initial poor treatment response. The main reason for adding platinum salts to neoadjuvant chemotherapy was to increase the likelihood of pCR achievement. Standard supportive treatments included administration 5HT3 inhibitors, ranitidine, and dexamethasone. Toxicity related to NACT was evaluated according to the National Cancer Institute’s Common Toxicity Criteria for Adverse Events scale (CTCAE) version 5.0 [45].

### 2.4. Early Clinical Response Evaluation and Definition of pCR

The initial assessment of disease extent was performed by clinical examination and imaging (ultrasound, mammography, and MRI in some cases). After the initial four cycles, the early clinical response (ECR) was evaluated by clinical examination and ultrasound imaging according to the Response Evaluation Criteria in Solid Tumor (RECIST) version 1.1 [46]. Patients with progressive disease (at least a 20% increase in the sum of diameters of target lesions) and stable disease were defined as non-responders, patients with complete response (the disappearance of all target lesions) or partial response (at least a 30% decrease in the sum of diameters of target lesions) were defined as responders. Tissue obtained from surgery after NACT was assessed by pathologists, and pCR was defined as the complete disappearance of all invasive carcinoma from breast and axillary lymphatic nodes, presence of in situ carcinoma was allowed (ypT0/is ypN0) [47]. The treatment design with the main evaluation points is shown in Figure 1.

### 2.5. Study Endpoints

The study aims were analyzed considering three endpoints: (i) ECR assessed during NACT, (ii) pCR assessed after surgery, and (iii) prognosis assessed by long-term outcomes, namely relapse-free survival (RFS) and overall survival (OS). RFS was calculated from the date of surgery to the date of the first event (locoregional relapse, distant relapse, or death from any cause), and OS was calculated from the date of surgery to the date of death from any cause. Patients without the observed event or lost from follow-up were censored at the date of the last appropriate visit.

### 2.6. Statistical Analysis

Patient and treatment characteristics were described using standard summary statistics, i.e., median and interquartile range (IQR) for continuous variables and frequencies and proportions for categorical variables. CEA, CA15-3, NLR, MLR, CAR, and SII levels were divided into two groups according to the optimal cut-off points selected by the criterion based on Youden’s index using receiver operating characteristic (ROC) curve analyses to discriminate between early responders and non-responders. The age was categorized into two groups using the cut-off value of 45 years derived based on the median. Treatment characteristics and numbers of adverse effects according to ECR or NACT type were compared using Fisher’s exact test or the Mann–Whitney test, as appropriate.

A univariable and multivariable logistic regression model was used to evaluate the association between clinical characteristics and ECR or pCR. Multivariable analysis was performed using backward stepwise selection based on the Akaike information criterion. The odds ratios (OR) and the 95% confidence intervals (CI) were estimated for each variable and optionally displayed using forest plots. The adjustment for ECR was used to assess pCR predictors for the whole group of patients. The type of NACT (platinum vs. nonplatinum) was considered as a stratification variable to assess pCR predictors. The significance of the interaction term (with the level of 0.1) was determined to identify the benefit of platinum salts addition for predictor variables. Survival probabilities were calculated using the Kaplan–Meier method. Survival curves were compared using the log-rank. The Cox proportional hazard model was used to calculate hazard ratios (HR). The follow-up was determined using the reverse Kaplan–Meier method. All statistical analyses were performed employing R version 4.0.3 [48] and a common significance level of 0.05.

## 3. Results

### 3.1. Baseline Characteristics and Early Clinical Response

The analyzed group included 237 retrospectively selected TNBC patients. The median age at diagnosis was 46 years (range 17–78 years). *BRCA1/2* mutation was detected in 72 (30%) patients, and 109 (46%) patients were found not to be carriers of the mutation. In 56 (24%) patients, the genetic testing was not indicated, and these patients were considered presumptive noncarriers (*BRCA* mutation undetected). The *BRCA* categorization details are described in Appendix A, including age distribution with respect to *BRCA* testing in Appendix A. Most patients had invasive breast carcinoma of no special type (IBC-NST) with a predomination of grade 3 tumors (195 patients, 75%). Before NACT, 177 (75%) patients were classified as cT1 or cT2. Lymph node involvement was present in 129 (54%) patients. The median value of the Ki-67 proliferative index was 76% (IQR 60–90%). Other patient and tumor pretreatment characteristics and laboratory parameters are summarized in Table 1.

ECR to NACT assessed after the initial four cycles was observed in 169 (71%) patients. ECR rate was significantly higher in patients with clinically smaller tumors (OR = 2.7, *p* = 0.002), unifocal type (OR = 4.0, *p* = 0.035), and on the borderline of significance with Ki-67 proliferative index ≥65% (OR = 1.8, *p* = 0.054). ECR was observed more frequently in *BRCA* mutated patients without reaching the statistical limit of significance (OR = 1.8, *p* = 0.072). In addition, the multivariable analysis indicated these characteristics as independent predictors for ECR (Table 1). The laboratory parameters were considered only in univariable analysis (Table 1) due to a greater proportion of unavailable data. High levels of CRP (cut-off 10 mg/L, OR = 0.17, *p* < 0.001), CAR (cut-off 0.095, OR = 0.40, *p* = 0.011), CA 15-3 (cut-off 18.4 kU/L, OR = 0.47, *p* = 0.020), and low levels of hemoglobin (cut-off 120 g/L, OR = 0.33, *p* = 0.032) were negatively associated with ECR.

### 3.2. Treatment Characteristics and Toxicity

Except for three patients, all patients started NACT with an A-based regimen; 35 (15%) patients received dose-dense AC. Administration of sequential NACT was indicated in 222 (94%) patients. NACT administration is summarized in Table 2. The addition of platinum salts to the standard NACT regimen was assigned in 68 (29%) patients, slightly more often in patients without ECR (*p* = 0.081). With regard to treatment management, patients with platinum NACT were younger, and 71% of them were *BRCA* carriers (vs. 14% for nonplatinum NACT). CBDCA was significantly more frequently administered in early clinical non-responders than CDDP (88% vs. 47%, *p* < 0.001). The statistically significant difference in either ECR or pCR rate between standard dosing regimens and dose-dense regimens was not observed (*p* = 0.673 for ECR, *p* = 0.533 for pCR). Platinum salts were administered in 19 (54%) patients with dose-dense regimens.

A total of 62 (91%) and 106 (73%) patients with platinum and nonplatinum NACT, respectively, experienced toxicity events of any grade (Table 3). This statistically significant difference in toxicity (*p* = 0.003) was not observed for serious adverse events of grade 3/4 (*p* = 0.155).

### 3.3. Pathological Complete Response and Effect of Platinum Salts Adding

The total pCR rate was 41%, specifically ypT0/ypN0 in 86 (36%) and ypTis/ypN0 in 11 (5%) patients. Achievement of pCR was strongly associated with ECR. Among 68 non-responders, only five (7%) patients achieved pCR. On the other side, pCR rate in responders was 54% (OR = 15.1, *p* < 0.001). The ECR-adjusted univariable analysis showed a higher pCR rate after platinum adding in all analyzed subgroups. The overall adjusted odds ratio for platinum NACT was 3.1. The significant benefit of platinum adding was observed in older (≥45 years) patients (*p*-value for interaction 0.015), patients with baseline cT3/cT4 (*p*-value for interaction 0.074), and with Ki-67 index ≥65% (*p*-value for interaction 0.035). Moreover, age ≥45 years was a marginally negative predictor of pCR for patients with nonplatinum NACT but positive for platinum NACT. Higher proliferation was a positive predictor of pCR for platinum NACT. Tumor size was a negative predictor of pCR for nonplatinum NACT. The ECR-adjusted univariable analysis is summarized in Figure 2 and Appendix A.

The subgroup of responders was analyzed separately for the practical reason and possible differences in sensitivity to NACT between early responders and non-responders (Figure 3 and Appendix A). A positive effect of platinum salts adding was observed in responders older than 45 years (*p*-value for interaction 0.014), baseline cT3/cT4 tumors (*p*-value for interaction 0.082), tumors with higher proliferative index (*p*-value for interaction 0.014), and in responders with baseline nodal involvement that was stable (according to cN staging) after the first four cycles of NACT. Multivariable analysis for early responders showed age, Ki-67 proliferative index and early change of cT and cN as independent predictors of pCR, taking into account the platinum salts adding. A significant benefit from platinum NACT, according to multivariable analysis, had responders older than 45 years with Ki-67 ≥ 65% who did not respond according to cN after the first four cycles of NACT (Figure 3 and Appendix A). Moreover, smaller tumors sensitive to the initial cycles of standard NACT were a positive predictor to achieve pCR.

### 3.4. Survival Outcomes

During a median follow-up period after surgery of 36 months (95% CI 27–41), 59 (25%) relapses, and 51 (22%) deaths were observed. Early clinical responders had statistically significantly better both OS (HR = 0.28, *p* < 0.001) and RFS (HR = 0.26, *p* < 0.001). The estimated five-year OS and RFS were 81.2% (95% CI 74.0–89.0%) and 78.4% (95% CI 71.3–86.2%) for responders and 44.3% (95% CI 31.4–44.3%) and 38.9% (95% CI 25.5–59.4%) for non-responders, respectively (Figure 4A,B). In the subgroup of early responders, patients achieving pCR had a statistically significantly lower risk of death (HR = 0.32, *p* = 0.011, Figure 4C) and relapse (HR = 0.17, *p* < 0.001, Figure 4D). Patients without ECR after the first four cycles NACT did not mostly achieve pCR. However, early non-responders supplemented with platinum salts in consequent cycles had better OS and RFS without reaching statistical significance (Figure 4C,D).

According to multivariable analyses (Table 4), ECR and pCR were independently associated with better OS and RFS. The type of NACT included in these analyses did not reach statistical significance. Nevertheless, platinum salts supplementation contributed to favorable survival outcomes, especially for OS.

## 4. Discussion

This retrospective analysis of neoadjuvant chemotherapy in TNBC patients following international guidelines was instigated by clinical practice and the need to identify patients who benefit most from the addition of platinum salts to the standard NACT regimens. The consequence of the presented comprehensive analysis is a suggestion for the decision-making process in platinum salts administration. ECR evaluation during NACT proved desirable. The responders profited from platinum regimens in the case of age above 45 years, very high proliferative tumors (Ki-67 ≥ 65%), and nodal involvement stable when evaluating ECR, regardless of *BRCA* status. The non-responders mainly did not achieve pCR. Nevertheless, more favorable survival outcomes were observed when platinum salts were administered.

The achievement of pCR was shown to be a predictor to have superior survival outcomes [17,18,19,20,21]. Strive for the complete disappearance of invasive carcinoma from breast and axillary lymphatic nodes by appropriate choice of NACT regimens was an endpoint of many prospective trials [49,50]. Furthermore, it was found that response evaluation during the NACT period could contribute by switching or modification of NACT to improve patients’ outcomes [24,51,52].

In this study, clinical evaluation of response during NACT was associated with pCR and also more favorable both OS and RFS (*p* < 0.001). The ECR evaluation was performed after four cycles of NACT, which coincide with the time of selecting considered and specific regimens (e.g., the addition of platinum salts). Previously published prospective studies evaluated the ECR after two cycles [24,51,52]. This study proved *BRCA* mutation, cT1/cT2 unifocal tumors, and Ki-67 ≥ 65% as independent predictors of ECR. The predictive role of these markers was evaluated more often with regard to pCR achievement than ECR [53,54]. Hong et al. retrospectively evaluated different characteristics of responders and non-responders [26]. Unlike the presented study, patients with different molecular subtypes were included, differences in cT were not found, and the proliferative index was not evaluated.

Abnormal levels of CRP, CAR, CA 15-3, and hemoglobin that may indicate locally advanced disease with a worse prognosis were associated with the worse initial response to NACT. Some previously published studies described the influence of various laboratory parameters on pCR achievement [55,56]. Considering the retrospective design of this study, additional investigation is necessary to prove the independent predictive role of laboratory markers.

The overall pCR rate of 41% observed in this study and the difference in pCR achievement between early responders and non-responders is concordant with published studies [26,29,57]. In this study, a positive influence of platinum salts on pCR was observed in all analyzed subgroups. Patients with ECR seem to be generally sensitive to NACT with a higher likelihood of pCR achieving. Modification of the regimen in consequent cycles may not be necessary for this subgroup. However, this study found that the addition of platinum salts to standard NACT increased the odds of pCR in patients above 45 years, Ki-67 ≥ 65%, and persisted nodal involvement regardless of *BRCA* status. Comparable findings taking into account the effect of initial NACT cycles influencing the next regimen setting were not previously published to our best knowledge. The predictive role of Ki-67 was evaluated in several studies, and its initial value may improve the prediction of treatment response [53,58,59]. Gamucci et al. published results of the retrospective study presenting the Ki-67 index with a cut-off of 50% as a positive predictor of pCR [57]. The benefit of adding platinum salts observed in patients above 45 years may be associated with the biological behavior of breast tumors in older women (e.g., menopausal status) and less sensitivity to NACT in general [54,60]. Therefore, these patients may need more intensive treatment to achieve a similar treatment outcome as younger patients.

The subgroup of early non-responders showed a low probability of pCR achievement. A separate evaluation of pCR predictors for this unfavorable subgroup could also be useful, but it was impossible to perform within this study. Nevertheless, it seems that the appropriate choice of NACT is possible to prolong time to death or relapse also in those patients who do not achieve pCR. The consequence could be considering modifying standard regimens from the beginning NACT for the group of patients in whom ECR is not expected.

NACT regimens administration followed international guidelines [13,14,36]. Contemporary sequential neoadjuvant regimens were administered in 94% of patients. However, a substantial impact of outdated regimens inclusion on performed analyses was not observed. Association of recently increasingly administered dose-dense regimens were not observed either with ECR or pCR. The impact on DFS and OS parameters in TNBC patients using dose-dense regimens in neoadjuvant settings is not clearly defined [61,62].

The administration of platinum salts was indicated according to physicians’ choice taking into account also *BRCA* status. Some trials evaluated the role of a platinum derivative in the context of the *BRCA* mutation [27,28,31,63]. Consistent with the conclusions of these studies, platinum salts to standard regimens were more often administered to *BRCA* mutated patients. Although this study’s results may be affected by this treatment management, according to ECR-adjusted analysis, there is no evidence of *BRCA* mutation’s influence on effectivity platinum regimens. These results are in concordance with some previously published results. Arun et al. evaluated *BRCA* status as an independent predictor for higher pCR rates in breast cancer patients [64]. According to the secondary analysis of the GeparSixto trial [65], *BRCA* carriers showed superior response rates without additive effects observed for carboplatin. Another published meta-analysis evaluating the role of platinum salts in *BRCA* mutated TNBC patients showed that adding platinum increases pCR rate without statistical significance [66]. Recently published prospective trial NeoSTOP evaluated carboplatin-containing NACT presented a higher pCR rate in *BRCA* mutated TNBC patients on the borderline of significance, but this study included only platinum-based regimens [67].

The use of platinum-based regimens or platinum salts adding to the standard regimens is often discussed in the context of higher incidence of toxicity and adverse events [68]. In the GeparSixto trial, the addition of carboplatin was associated with a higher rate of hematological and non-hematological toxicity, and the dose of carboplatin was reduced from AUC 2.0 to AUC 1.5 [29]. On the other hand, a more favorable toxicity profile was observed for the AC-free carboplatin regimen in the NeoSTOP trial [67]. According to our study results, toxicity was more common in patients with platinum NACT but especially lower grades. A statistically significant difference in grade 3/4 toxicity rate was not observed. A retrospective nature, differences in chemotherapy regimens (cisplatin vs. carboplatin), and cytostatics dosage might explain these different results.

The achieving of ECR and pCR was strongly associated with favorable RFS and OS parameters and proved independent predictors in multivariable analyses. The influence of platinum salts adding on survival outcomes was not generally observed, similarly as previously published [69]. However, from the retrospective nature of the study and the fact that the administration of platinum salts was not independent of ECR, the effect of platinum regimens is necessary to evaluate in the context of the initial response status. The multivariable analysis proved the non-significantly superior effect of NACT type on OS and RFS.

We are aware of some limitations of the study caused mainly by its retrospective nature. Firstly, neoadjuvant treatment regimens and cycles were at the physician’s discretion. Moreover, *BRCA* status was unknown in some patients during the NACT, or genetic testing was not performed. The joint classification of *BRCA* noncarriers and untested patients was carefully considered and used. Untested patients were mainly elderly patients, and the expected prevalence of *BRCA* germline mutations in this group of TNBC patients is low [70]. Further, the cut-off value of Ki-67 was arbitrarily chosen as 65%. This choice was considering Ki-67 distribution in TNBC tumors [71] and multiple endpoints in this study. Finally, the patient number was not so large, especially for subgroup analyses. On the other hand, the reported data reflect the real neoadjuvant treatment management of a single center and directly impact clinical practice.

As already mentioned, some results may have immediate use in clinical practice, although they may be influenced by retrospective design or small sample size. Therefore, it is appropriate to verify questionable findings in the prospective design. In particular, the subgroup of non-responders deserves more attention, considering the potential modification of standard NACT regimens and their timing.

## 5. Conclusions

Clinical response evaluation during NACT is useful to identify patients with favorable long-term outcomes (OS or RFS). The achievement of pCR in early responders is an essential target that may be directed by a controlled modification of NACT regimen. According to this study results, the platinum salts addition was beneficial in patients who achieved ECR with age ≥45 years, tumors with Ki-67 index ≥65%, and without early regression of nodal involvement. *BRCA* mutation was a predictor of ECR and showed the general sensitivity of *BRCA* carriers to NACT, but not specifically to platinum salts in this study. A low probability of pCR achievement was observed in early non-responders but there were more favorable survival outcomes when platinum salts were added.

## Figures and Tables

**Figure 1 cancers-13-01586-f001:**
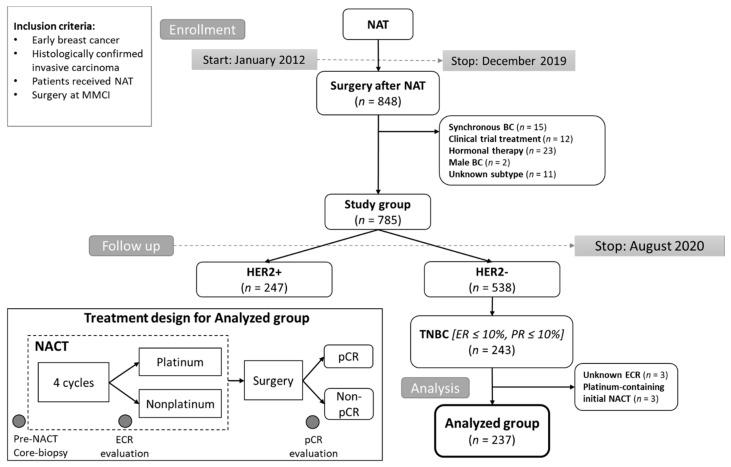
CONSORT diagram of the study population and diagram of treatment design for the analyzed group. Abbreviations: MMCI, Masaryk Memorial Cancer Institute; NAT, neoadjuvant therapy; NACT, neoadjuvant chemotherapy; TNBC, triple-negative breast cancer; BC, breast cancer; ER, estrogen receptor; PR, progesterone receptor; HER2, human epidermal growth factor receptor; ECR, early clinical response; pCR, pathological complete response.

**Figure 2 cancers-13-01586-f002:**
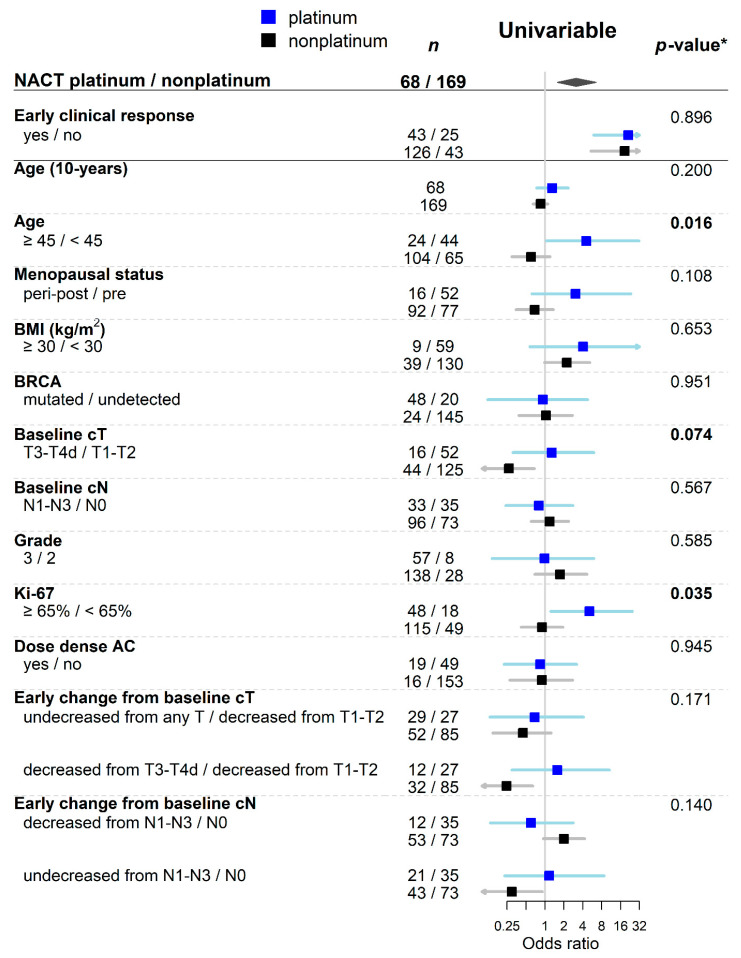
Forest plot of ECR-adjusted univariable analysis of pCR predictors showing odds ratios with confidence intervals according to NACT type. * *p*-value for the interaction term between NACT type and predictor. Abbreviations: NACT, neoadjuvant chemotherapy; BMI, Body mass index.

**Figure 3 cancers-13-01586-f003:**
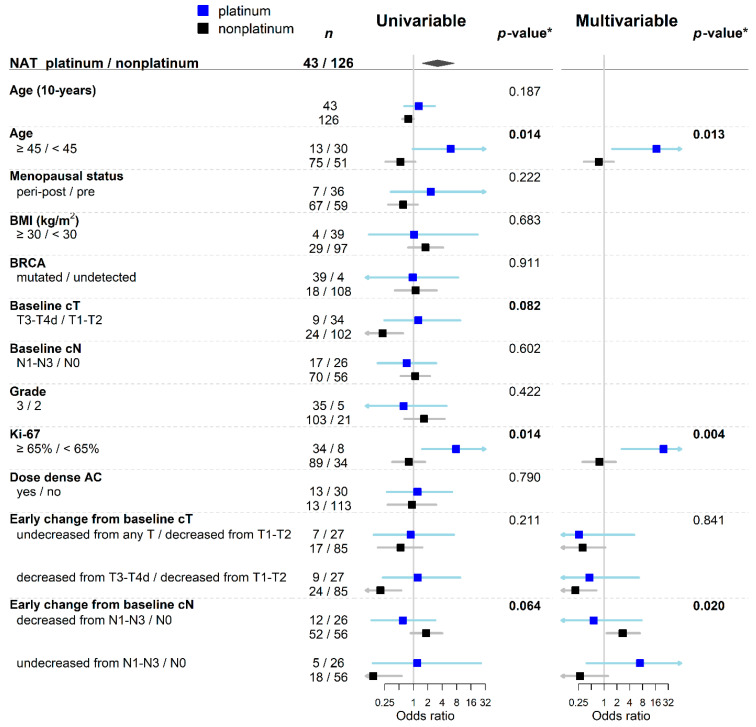
Forest plot of univariable and multivariable analysis of pCR predictors showing odds ratios with confidence intervals according to NACT type for early responders. * *p*-value for the interaction term between NACT type and predictor. Abbreviations: NACT, neoadjuvant chemotherapy; BMI, Body mass index.

**Figure 4 cancers-13-01586-f004:**
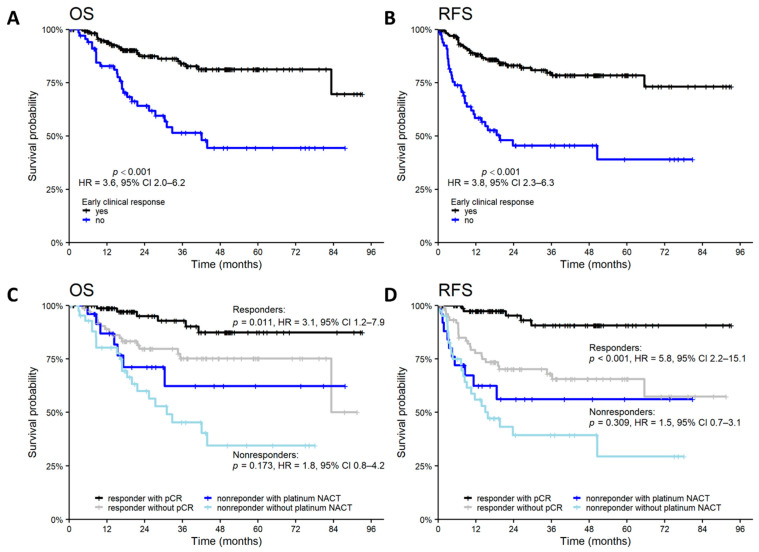
Kaplan-Meier estimates according to early clinical response for (**A**) overall survival and (**B**) relapse-free survival. Kaplan-Meier estimates for responders according to pCR achievement and nonresponders according to the type of NACT in terms of (**C**) overall survival and (**D**) relapse-free survival. Abbreviations: pCR, pathological complete response; NACT, neoadjuvant chemotherapy; OS, overall survival; RFS, relapse-free survival; HR, hazard ratio; CI, confidence interval.

**Table 1 cancers-13-01586-t001:** Baseline patient and tumor characteristics. Univariable and multivariable analyses of early clinical response predictors.

Variables	Values	Overall	Early Clinical Response		Univariable Analysis			Multivariable Analysis		
		*n* = 237	No *n* = 68	Yes *n* = 169	OR	95% CI	*p*-Value	OR	95% CI	*p*-Value
Age (years)	Median (IQR)	46 (37, 58)	49 (39, 59)	45 (37, 57)	0.98	0.96, 1.01	0.140			
	Range	17, 78	23, 77	17, 78						
Age	<45	109 (46%)	28 (26%)	81 (74%)	—	—	0.344			
	≥45	128 (54%)	40 (31%)	88 (69%)	0.76	0.43, 1.34				
Menopausal status	pre	129 (54%)	34 (26%)	95 (74%)	—	—	0.386			
	peri-post	108 (46%)	34 (31%)	74 (69%)	0.78	0.44, 1.37				
BMI (kg/m²)	<30	189 (80%)	53 (28%)	136 (72%)	—	—	0.663			
	≥30	48 (20%)	15 (31%)	33 (69%)	0.86	0.44, 1.74				
BRCA1/2	undetected ^1^	165 (70%)	53 (32%)	112 (68%)	—	—	0.072	—	—	0.152
	mutated	72 (30%)	15 (21%)	57 (79%)	1.80	0.95, 3.56		1.63	0.84, 3.33	
Baseline cT	T1-T2	177 (75%)	41 (23%)	136 (77%)	—	—	**0.002**	—	—	**0.011**
	T3-T4d	60 (25%)	27 (45%)	33 (55%)	0.37	0.20, 0.68		0.43	0.22, 0.82	
Focality	multi-centric/focal	10 (4%)	6 (60%)	4 (40%)	—	—	**0.035**	—	—	0.090
	unifocal	227 (96%)	62 (27%)	165 (73%)	3.99	1.10, 16.1		3.44	0.82, 15.4	
Baseline cN	N0	108 (46%)	26 (24%)	82 (76%)	—	—	0.149			
	N1–N3	129 (54%)	42 (33%)	87 (67%)	0.66	0.37, 1.16				
Grade	2	36 (16%)	10 (28%)	26 (72%)	—	—	0.859			
	3	195 (84%)	57 (29%)	138 (71%)	0.93	0.41, 2.00				
	Unknown	6	1	5						
Ki-67 (%)	Median (IQR)	76 (60, 90)	74 (52, 90)	77 (64, 90)	1.01	0.99, 1.02	0.386			
	Range	15, 100	38, 98	15, 100						
	Unknown	7	3	4						
Ki-67	<65%	67 (29%)	25 (37%)	42 (63%)	—	—	0.054	—	—	0.079
	≥65%	163 (71%)	40 (25%)	123 (75%)	1.83	0.99, 3.37		1.77	0.94, 3.34	
	Unknown	7	3	4						
Histology	IBC-NST	226 (95%)	64 (28%)	162 (72%)						
	Other	11 (4.6%)	4 (36%)	7 (64%)						
Laboratory parameters *										
LDH (μkat/L)	<3.8	170 (80%)	46 (27%)	124 (73%)	—	—	0.275			
	≥3.8	42 (20%)	15 (36%)	27 (64%)	0.67	0.33, 1.39				
CRP (mg/L)	<10	171 (90%)	43 (25%)	128 (75%)	—	—	**<0.001**			
	≥10	18 (9.5%)	12 (67%)	6 (33%)	0.17	0.06, 0.46				
CAR	<0.095	144 (76%)	35 (24%)	109 (76%)	—	—	**0.011**			
	≥0.095	45 (24%)	20 (44%)	25 (56%)	0.40	0.20, 0.81				
CEA (μg/L)	<2.8	176 (86%)	46 (26%)	130 (74%)	—	—	0.101			
	≥2.8	29 (14%)	12 (41%)	17 (59%)	0.50	0.22, 1.15				
CA 15-3 (kU/L)	<18.4	86 (43%)	17 (20%)	69 (80%)	—	—	**0.020**			
	≥18.4	116 (57%)	40 (34%)	76 (66%)	0.47	0.24, 0.89				
Hemoglobin (g/L)	<120	17 (8.0%)	9 (53%)	8 (47%)	—	—	**0.032**			
	≥120	196 (92%)	53 (27%)	143 (73%)	3.04	1.11, 8.48				
LMR	<5.53	197 (92%)	55 (28%)	142 (72%)	—	—	0.195			
	≥5.53	16 (7.5%)	7 (44%)	9 (56%)	0.50	0.18, 1.45				
NLR	<2.58	129 (61%)	35 (27%)	94 (73%)	—	—	0.433			
	≥2.58	84 (39%)	27 (32%)	57 (68%)	0.79	0.43, 1.44				
SII	<774	151 (71%)	39 (26%)	112 (74%)	—	—	0.104			
	≥774	62 (29%)	23 (37%)	39 (63%)	0.59	0.31, 1.12				

^1^ Undetected includes patients without *BRCA1/2* germline mutation or not tested. * Laboratory parameters were unavailable for some patients: LDH, *n* = 25; CRP and CAR, *n* = 48; CEA, N = 32; CA 15-3, *n* = 35; Hemoglobin, LMR, NLR, and SII, *n* = 24. Cut-off values for CEA, CA15-3, NLR, MLR, CAR, and SII were determined using ROC analysis. Cut-off values for LDH, CRP and Hemoglobin were determined according to lower/upper reference limit. Abbreviations: OR, odds ratio; IQR, interquartile range; CI, confidence interval; BMI, Body mass index; IBC-NST, invasive breast carcinoma of no special type; LDH, lactate dehydrogenase; CRP, C-reactive protein; CAR, C-reactive protein to albumin ratio; NLR, neutrophil to lymphocyte ratio; LMR, lymphocyte to monocyte ratio; SII, systemic immune-inflammation index (neutrophil × platelet/lymphocyte). Bold: Highlight the statistically significant value on the level 0.05.

**Table 2 cancers-13-01586-t002:** Neoadjuvant chemotherapy characteristics.

Variables	Values	Overall	Early Clinical Response		
		*n* = 237	No *n* = 68	Yes *n* = 169	*p*-Value
Regimens of NACT	A-based only	12 (5.1%)	4 (5.9%)	8 (4.7%)	
	A → T	154 (65%)	37 (54%)	117 (69%)	
	T-based only	2 (0.8%)	1 (1.5%)	1 (0.6%)	
	CMF	1 (0.4%)	1 (1.5%)	0 (0%)	
	A → T + CBDCA	42 (18%)	22 (32%)	20 (12%)	
	A → CDDP	26 (11%)	3 (4.4%)	23 (14%)	
Dose dense AC		35 (15%)	9 (13%)	26 (15%)	0.673
NACT	nonplatinum	169 (71%)	43 (63%)	126 (75%)	0.081
	platinum	68 (29%)	25 (37%)	43 (25%)	
Platinum salts	CBDCA	42 (62%)	22 (88%)	20 (47%)	**<0.001**
	CDDP	26 (38%)	3 (12%)	23 (53%)	
Time from diagnosis to NACT (days) ^1^	Median (IQR)	20 (14, 29)	20 (13, 32)	21 (14, 29)	0.500
	Range	0, 130	0, 113	2, 130	
	Unknown	18	2	16	
			NACT		
		Overall *n* = 237	Nonplatinum *n* = 169	Platinum *n* = 68	*p*-Value
Time from NACT to surgery (days) ^2^	Median (IQR)	30 (23, 38)	29 (22, 37)	33 (26, 41)	0.158
	Range	3, 159	3, 159	3, 81	
	Unknown	16	15	1	

^1^ Time from the diagnosis to the start of NACT; ^2^ Time from the termination of NACT to surgery. Abbreviations: NACT, neoadjuvant chemotherapy; A, anthracyclines (doxorubicin, epirubicin); T, taxanes (paclitaxel, docetaxel); CDDP, cisplatin; CBDCA, carboplatin; CMF, cyclophosphamide, methotrexate, 5-fluorouracil; IQR, interquartile range. Bold: Highlight the statistically significant value on the level 0.05.

**Table 3 cancers-13-01586-t003:** Hematological and non-hematological adverse events according to NACT type.

Toxicity	Nonplatinum NACT *n* = 145 *		Platinum NACT *n* = 68		*p*-Value Any Grade	*p*-Value Grade 3–4
	Grade 1–2	Grade 3–4	Grade 1–2	Grade 3–4		
Overall toxicity	38 (26%)	68 (47%)	23 (34%)	39 (57%)	**0.003**	0.155
Myelotoxicity	19 (13%)	54 (37%)	16 (24%)	32 (47%)	**0.005**	0.173
Leukopenia/Neutropenia	17 (12%)	52 (36%)	13 (19%)	31 (46%)		
Anaemia	3 (2.1%)	0 (0%)	3 (4.4%)	2 (2.9%)		
Thrombocytopenia	1 (0.7%)	1 (0.7%)	2 (2.9%)	0 (0%)		
Febrile neutropenia	0 (0%)	2 (1.4%)	0 (0%)	1 (1.5%)		
Nonhaematological toxicity	67 (46%)	23 (16%)	36 (53%)	11 (16%)	0.317	0.953
Skin and mucosal toxicity	19 (13%)	5 (3.4%)	2 (2.9%)	3 (4.4%)		
Nausea, Vomiting	21 (14%)	10 (6.9%)	20 (29%)	4 (5.9%)		
Diarrhoea	4 (2.8%)	0 (0%)	2 (2.9%)	0 (0%)		
Neurotoxicity	29 (20%)	9 (6.2%)	14 (21%)	4 (5.9%)		
Hepatotoxicity	0 (0%)	0 (0%)	1 (1.5%)	0 (0%)		
Premature Termination	19 (13%)		9 (13%)			

Abbreviations: NACT, neoadjuvant chemotherapy. * Toxicity was unavailable for 24 patients treated by nonplatinum NACT outside MMCI. Bold: Highlight the statistically significant value on the level 0.05.

**Table 4 cancers-13-01586-t004:** Multivariable analysis for OS and RFS.

Variables	Values	OS			RFS		
		HR	95% CI	*p*-Value	HR	95% CI	*p*-Value
ECR	no/yes	2.66	1.47, 4.83	**0.001**	2.43	1.43, 4.13	**0.001**
NACT	platinum/nonplatinum	0.58	0.29, 1.19	0.139	0.74	0.40, 1.36	0.331
pCR	no/yes	3.36	1.36, 8.28	**0.008**	6.47	2.51, 16.7	**<0.001**

Abbreviations: ECR, early clinical response; NACT, neoadjuvant chemotherapy; pCR, pathological complete response; OS, overall survival; RFS, relapse-free survival; HR, hazard ratio; CI, confidence interval. Bold: Highlight the statistically significant value on the level 0.05.

## Data Availability

The data presented in this study are available on request from the corresponding author.

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
