# Peer review of "Neoadjuvant Chemotherapy of Triple-Negative Breast Cancer: Evaluation of Early Clinical Response, Pathological Complete Response Rates, and Addition of Platinum Salts Benefit Based on Real-World Evidence"

_cancers, 2021, doi:10.3390/cancers13071586_

Round 1

Reviewer 1 Report

The authors have addressed the comments very well. 

Author Response

Thank you for the review.

Reviewer 2 Report

The purpose of this retrospective study is to find out the influence of palatium salts on the higher pCR and survival rates

to standard chemo regimen in TNBC patients.

Following the revision, the MS is improved a lot and really a meaningful observations for breast cancer clinicians. Very recently Priyanka Sharma et al

published the similar report with a prospective study by adding carboplatin to anthracycline-based chemotherapy (NeoSTOP study) (please see CCR 202; 27:975-82).

Authors may incorporate this reference in their discussion part.

Please also correct the small typo error in line 14 of page 8 where they mentioned Table 2 but in line 23 in the actual table it is Table 1.

Author Response

Thank you for the review and the suggestion of the recent reference for Discussion. The Discussion has been modified. The typo error was caused by MDPI office during the submission process, but it has been checked and corrected.

This manuscript is a resubmission of an earlier submission. The following is a list of the peer review reports and author responses from that submission.

Round 1

Reviewer 1 Report

The authors present a well-written retrospective study of 237 triple-negative breast cancer patients with a median follow-up of 36 months and evaluated the role of adding platinum salts into standard neoadjuvant chemotherapy (NACT). They show that BRCA mutation, smaller unifocal tumors and Ki-67 ≥ 65% were independent predictors of early clinical response (ECR). In addition, they show in multivariate analysis that platinum-containing NACT was observed in early responders ≥ 45 years, Ki-67 ≥ 65% and persisting lymph node involvement regardless of BRCA status. ECR was associated with better survival. However, toxicity was increased in platinum-containing NACT.

These real-world data are interesting and well-presented, however, several crucial questions should be addressed by the authors:

  • Why did the authors decided to give 4 cycles of NACT and not 2 as used in other publications to define ECR?
  • Was there a difference in the significance of ECT between sequential (e.g. EC – Pw) abd combined regimens (e.g. FEC).
  • Why did the consider the 24% TNBC without BRCA testing as presumptive noncarriers. Would it change their results if these patients were not included in their analyses?
  • Why was the cut-off of 65% for Ki-67 selected? How did the median for Ki-67 perform?
  • Some oft he regimens investigated are rather outdated (e.g. CMF or anthracycline only). It might be interesting and strengthen the findings to present results of patients treated with a contemporary sequential neoadjuvant chemotherapy ( A → T or A → T +CBDCA).
  • Was the significant impact of ECS on survival independent from pCR in multivariate analysis?
  • Did the authors notice a significant effect of platinum salts on survival regardless oft he initial response status?

Reviewer 2 Report

Article contains good amount of data but the design as well as results are not novel and very much predicted too. In ASCO guidelines clearly mentioned that "patients with TNBC

who have clinically node-positive and/or at least T1C disease should be offered an anthracycline- and taxane-containing regimen; those with cT1a or cT1bN0 should not

routinely offered neoadjuvant therapy. Carboplatin may be offered (Physician's choice) to patients with TNBC to increase pCR (PMID:33507815)". Hence it is not a novel study

design and results are very much predicted.

Critical Comments:

  1. Authors not mentioned everywhere that that patients are node positive or not
  2. Authors failed to mentioned about BRCA's mutation status related to somatic or germline
  3. In the text authors mentioned patients number in NACT #30 (line 12 in the section of 3.2) and in the table #2, they mentioned patients #35
  4. Authors have any suggestion between dose dense AC vs platinum salts...
  5. In section 3.5, line 8, nonresponders are with stable disease or not
  6. In Figure #4, how come HR is high in responder than non-responders ….please explain
  7. Discussion is very poorly done. Needs total rewriting.

Reviewer 3 Report

The manuscript entitled "Neoadjuvant Chemotherapy of Triple-Negative Breast Cancer: Evaluation of Early Clinical Response, Pathological Complete Response Rates, and Addition of Platinum Salts Benefit Based on Real-World Evidence" by Holanek et al. describes about the use of Early Clinical response for the treatment of triple negative breast cancer using platinum salts as neoadjuvant chemotherapy. The authors used 237 patients for NACT and concluded that ECR can be used for NACT using platinum drugs irrespective of BRCA status. The study is thoroughly analyzed and described. Though the patient samples are relatively smaller, it is good initial study for the use of ECR for NACT using platinum drugs. I highly recommend for the publication of this study.